# Psychometric Properties of the Polish Version of the Parental Feelings Inventory

**DOI:** 10.3390/ijerph18116003

**Published:** 2021-06-03

**Authors:** Dominika Fijałkowska, Eleonora Bielawska-Batorowicz

**Affiliations:** Faculty of Educational Sciences, Institute of Psychology, University of Lodz, 91-433 Lodz, Poland; eleonora.batorowicz@now.uni.lodz.pl

**Keywords:** emotions, parental feelings, parental role, parenting behaviors, parent–child relations

## Abstract

Previous reports from international literature have emphasized the specific nature of the feelings experienced by parents related to their role, differentiating them from other emotional states. Researchers highlighted the role of parental emotions in parent–child interaction and child development. The aim of the present study was to create the Polish version of the Parental Feelings Inventory (PFI), which is a rating scale designed to determine parental emotions within the parenting role, and to assess its psychometric properties. The participants included 191 parents (102 mothers and 89 fathers) aged 22 to 55 years (*M* = 37.18, *SD* = 6.85), both parents of healthy children and those of children with chronic diseases or disabilities. All participants completed the translated PFI questionnaire, Parental Attitude Scale (SPR), SUPIN S20 and SUPIN C20. The results support a three-factor solution (*Angry*, *Happy*, and *Anxious*/*Sad*) and a structure of 22 items for the PFI. The internal consistency for the complete scale was *α* = 0.78 for mothers and *α* = 0.76 for fathers. Cronbach’s *α* coefficients for individual factors for the Polish version were good: all above 0.80. The Polish PFI correlated with measures of parental attitudes and positive/negative affect; in addition, the Polish version confirms the basic psychometric criteria of the original and yields the same results. Therefore, the Polish version of the PFI is a valid and reliable tool for measuring parental feelings and can be successfully used as a measure of emotional experiences in the parenting role.

## 1. Introduction

While emotions have traditionally been examined as stable trait-like characteristics, recent research has focused more on their dynamics and variability, both among individuals and across contexts [1]. Emotion varies or fluctuates in response to momentary events or stimuli such as social interactions or stress [1,2]. Given that parenting presents a variety of environmental stimuli and stressors [3], understanding how emotions may vary across caregiving and non-caregiving contexts is an important area of inquiry. Although parenthood is a nearly universal experience, the influence that caring for children has on emotional experiences remains poorly understood [4]. In addition, little is known about how feelings occurring during parenthood are shaped by the context in which parenting takes place [5]. Moreover, as parents are responsible for managing both their own emotions and those of their children, parental emotions are complex and have an effect on the well-being of both parent and child [6]. Eisenberg and colleagues’ [7] model of parental socialization of emotion gives primary significance to the impact of parents’ emotional expressions and behaviors on their children’s development of emotional competence and well-being. The model suggests that parents’ emotion socialization behaviors (ESBs), particularly their expression of emotions and their reactions to children’s emotions, have a direct impact on children’s emotional arousal and learning about emotions. The model also indicates other processes influencing emotion socialization, assigning a particularly important role (as both a predictor of ESBs and a moderator of the relation between ESBs and socialization) to the parenting style (e.g., warmth, hostility, permissiveness, etc.) [7,8].

Most of the research on parental emotions focuses on the overall levels of emotions or their retrospective descriptions [9]. However, recent studies have turned more to examining the parents’ experiences of emotions in real time, with a particular focus on their intensity: it has been found that mothers tend to experience higher mean levels, or intensities, of positive emotions when caring or interacting with their children than when they are not [10,11]. Research also emphasizes the specific nature of parental emotions: parents may actually experience heightened levels of both positive and negative emotions when caring for their children compared to when they are not [4], mothers have reported higher levels of both positive and negative emotions while spending time with their children [12], both mothers and fathers have reported more anger, stress, and anxiety than nonparents [13], and parenting activities were found to elicit more negative emotion than any other activity besides work [14]. Rueger et al. [15] report that the specific parental emotional states are related to parenting behavior: positive emotions predict adaptive, supportive parenting while negative parental emotions predict maladaptive parenting. Numerous studies [15,16] highlight the role of negative emotions in the parent–child relationship, linking parent psychopathology to parenting and child functioning.

As Bradley et al. [17] indicate, parenthood is a multi-faceted state, and hence, parents can experience different emotions while in different aspects of the parental role. Nelson et al. [9] report that parents experiencing positive emotions have a stronger sense of social roles and a greater sense of meaning in life, being happier and appearing to better experience the pleasure of parenting; in contrast, parents experiencing negative emotions or financial and marital relationship difficulties tend to be unhappy. Leerkes et al. [18] suggest that it is particularly important to study parental emotions during challenging parenting situations because parenting behavior in the context of child-related stress is uniquely associated with child outcomes. It is known that parents of children with disabilities experience more negative emotional states than those of children without [19]. A study by Tekinarslan [20] indicated that the mothers of children diagnosed with autism spectrum disorders (ASD) had experienced grief, denial, anxiety and fear after the diagnosis and that these feelings had not changed with time.

As noted by Bradley et al. [17], the literature in this area is dominated by the measurement of psychopathological symptoms in parents. A review of existing adjective checklists allowed Bradley et al. to conclude that none of them assess feelings specific to the parenting role. However, in order to more fully understand parental feelings, it is necessary to create a more comprehensive tool. Global measures of mood may only partially capture the emotions experienced within the parenting context. These emotional experiences, i.e., those related to parental roles, seem to play key roles in parent–child interaction and child development. However, no tool had been developed that could directly measure parental emotions. In response, Bradley et al. [17] designed the Parental Feelings Inventory (PFI) to fill this gap using available measurement methods. The PFI itself was designed following an analysis of the differences in the meanings of the terms *emotions*, *mood*, and *affect* in the available literature [21,22]. Thus, it focuses on the specific emotions that mothers and fathers feel within their role as parents.

In Poland, researchers have been expanding knowledge of the psychology of emotions and of family psychology for many years. Within Polish literature, a significant number of studies have been performed on emotional states [23,24,25,26] and also on the functioning of families [27,28,29], and these highlight the need to verify, and broaden, current knowledge on these topics. Within these two disciplines, research methods are also being developed separately [30,31]. However, although these two areas of psychology have traditionally been considered separately, the specific nature of the feelings experienced by parents related to their role has been emphasized in international literature (mentioned above). Parents respond to a variety of situations related to their role expressing their feelings, both positive and negative. The measures that reflect emotions outside any specific context might not be sufficient to grasp properly parental emotional reactions and feelings. Therefore, there is a clear need to provide an appropriate measure that can combine these two widely discussed areas while taking into account the specificity of the parental situation. In research, the use of an appropriate measure is vital for data reliability, while in the clinical setting, it helps to get the involvement and cooperation of potential clients. Such a measure was so far not available for studies in the population of Polish parents.

To fill this gap, in line with previous international studies, we decided that the Parental Feelings Inventory (PFI) is a suitable candidate for adaptation to the context of the Polish population, being a tool with high psychometric properties that can accurately reflect the feelings of parents associated with their parental role. Hence, the aim of our study was to translate, adapt, and assess the psychometric properties of the Polish version of the PFI questionnaire. In addition, we intended to examine the concurrent validity of the Polish version of the PFI by comparing it with tools available in Polish literature that also measure positive/negative affect and parental attitudes. The results of such a procedure will support the use of the Polish version of the PFI as the tool that can replace other measures while providing the equally reliable data and being more appropriate for the context of the parental role.

Following from international studies indicating that parents of children with disabilities tend to display different emotions to those of children without a diagnosed disability [19,20], we decided to examine whether this was also true for the two groups of parents from the Polish population. Many international studies confirm the hypothetical assumptions in this area. In case of disabled children, disability and cognitive problems bring extra needs and demands for the parents. As a result of continuous pressures, these parents develop stress, depression, and anxiety [32] more often than parents of non-disabled children [33]. Bitsika and Sharpley [34] found that 60 to 80% of parents of children with ASD had moderate-to-severe levels of depression, anxiety, and/or stress. Researchers have found that parents of children with ASD experience significantly higher levels of distress than parents of children with typical development, neurological disorders, learning disabilities, genetic, or chromosomal conditions or other developmental or behavioral disorders [35,36]. Similarly, in the study by Arzeen et al. [37], intellectually disabled children’s parents reported higher stress, depression, and anxiety than non-disabled children’s parents. According to such findings, we assumed that the two groups of parents (parents of children with disabilities and those of children without a diagnosed disability) would demonstrate significant differences in parental feelings.

In line with Watson et al. [38], who recognized two independent affective dimensions, we chose to examine whether positive or negative affect (two factors representing affective dimensions) would correlate with parental feelings (specific emotions that parents feel within their role as parents). Such analyses were not conducted so far. Therefore, it was assumed that PFI scores related to positive feelings would be positively associated with positive affect, while PFI scores related to negative feelings would be associated with negative affect.

In his typology of parental attitudes, Plopa [39] points out that the parental attitude contains three components (cognitive, emotional/motivational, and behavioral), of which the emotional is considered the most important. On the grounds of the emotional and behavioral component, six types of parental interaction with children were distinguished. These include one described as a continuum (acceptance–rejection), as well as autonomous, overly demanding, inconsistent, and excessively protective attitudes. According to this approach to parental attitudes [39], accepting and autonomous attitudes are treated as attitudes of positive and desired significance, while overly demanding, inconsistent, and excessively protective attitudes are treated as those of negative or undesirable significance. It can be hypothesized that positive parental attitudes would be linked to the expression of positive parental feelings. To confirm such an assumption, the present study will examine such relationships. More specifically, it is assumed that parental positive (accepting and autonomous) attitudes (through their positive emotional loading) are associated with PFI scores related to positive feelings, and that parental negative attitudes (through their negative emotional loading) are associated with PFI scores related to negative feelings.

## 2. Materials and Methods

### 2.1. Participants

The participants included 191 parents (102 mothers and 89 fathers) aged from 22 to 55 (*M* = 37.18, *SD* = 6.85) recruited through contacts with associations, foundations, schools, kindergartens, and friends. All participants had at least one child under the age of 18. We controlled the place of recruitment to make sure that younger or older children are not overrepresented in the sample. Both parents of healthy children and parents of children with chronic diseases or disabilities were invited to participate in the study. Children with disabilities included those with autism spectrum disorders (29), cerebral palsy (14), intellectual disability (6), and other types of health problems (14). Detailed characteristics of participants are presented in Table 1.

### 2.2. Measures

#### 2.2.1. PFI Parental Feelings Inventory

The PFI was designed by Bradley et al. [17] to assess parental emotions within the parenting role. Initially, the authors created a checklist of 31 adjectives associated with emotions; parents were asked to rate the degree to which they experienced each one during the past week in their role as a parent on a 7-point Likert scale (from 1 = “not at all” to 7 = “extremely”).

Following this initial stage, Bradley et al. [17] went on to evaluate the psychometric indicators of the questionnaire in a sample of parents (149 mothers and 107 fathers) of preschool-age children, finally arriving at a version consisting of a 26-item tool with a three-factor structure: *Angry* (6 items), *Happy* (11 items), and *Anxious*/*Sad* (9 items). The scale demonstrated good reliability (Cronbach’s alpha for total scale: *α* = 0.85 for mothers; *α* = 0.87 for fathers; all *α* values were above 0.90 for individual factors) and correlated, as theoretically expected, with other measures of parent and child functioning: parenting stress, depression, parenting style, and child behavior.

With the authors’ consent, this tool creation process was reproduced in the present study, and the initial version of 31 items was used as the basis for further translation and adaptation procedures. By reproducing this original process while adapting the tool, our aim is to compare the results obtained in the Polish population with the results of the original study.

#### 2.2.2. SPR Parental Attitude Scale

The SPR questionnaire by Plopa [39] is intended to measure parental attitudes. The questionnaire consists of two versions: one for maternal parental attitudes and the other for paternal attitudes. Each version contains 50 statements grouped in five scales (10 items in each scale) corresponding to five parental attitudes: acceptance–rejection, excessively demanding, autonomy, inconsistent, and excessively protective. The participant responds to each statement on a five-point scale. The raw results are obtained for each scale in the range of 10–50 points, and then, these are converted to sten results. The results expressed in stens are interpreted as the degree of intensity of a specific attitude. The scale has high internal consistency for both maternal and paternal versions: Cronbach’s alpha coefficients range from 0.73 to 0.89 [39].

#### 2.2.3. SUPIN Positive and Negative Affect Schedule

Negative and positive emotions were assessed with the Positive and Negative Affect Schedule (PANAS) [40]; Polish adaptation: Brzozowski [41]. For the Polish adaptation of the PANAS questionnaire, the name SUPIN is used. The tool measures the severity of negative and positive emotions with a list of adjectives. It has a total of four versions: two of them (one shorter 20 items, and one longer 30 items) are designed to measure current emotional states (S20 and S30), while the other two, of similar lengths, relate to relatively constant affective features (C20 and C30). The respondent assesses the extent to which the adjectives describe their current state (version S) or how they usually feel (versions C) on a scale of 1 to 5. The final result is calculated separately for the scale of positive feelings (PU) and the scale of negative feelings (NU). The internal consistency of the scales is high or satisfactory: Cronbach’s alpha coefficients range from 0.73 to 0.95 depending on the version and the type of sample. The C versions also have high absolute stability [41]. The present study used two shorter versions of the SUPIN, *viz.* S20 and C20, that maintain high psychometric indices, to check whether the adapted PFI questionnaire would be associated with emotions as a state or as a trait.

### 2.3. Procedure

After obtaining the consent of the author of the questionnaire, Professor Elizabeth Bradley, for the use of the tool in our research and to produce a Polish adaptation, the questionnaire was translated into Polish in accordance with the following procedure: the initial translation was performed by two independent translators; then, the text was assessed by a team consisting of three pairs of parents and psychologists; finally, a back translation was performed, and a decision was made on the final version.

Then, parents were invited to participate in the study. The principles of the study were presented to the participants through an invitation letter and by direct conversation. Parents who agreed to participate in the study signed informed consent forms (in accordance with the guidelines of the Research Bioethics Committee), which they placed in a separate box to ensure anonymity and data confidentiality. Then, they received an envelope containing a set of questionnaires consisting of a demographic survey, PFI, SPR, SUPIN S20, and SUPIN C20. They returned the completed questionnaires in the sealed envelopes, ensuring anonymity. After completing the entire study, envelopes were opened at random, and the sets of questionnaires were coded with ordinal numbers (randomly).

The data collected in this way were subjected to statistical analyses performed with the IBM Statistical Package for the Social Sciences (SPSS 25) (IBM Corporation, Armonk, NY, USA) and AMOS 25 (Amos Development Corporation, Wexford, US). The entire group of participants (N = 191) was divided randomly into two parts: the first (N = 96) for the exploratory factor analysis and the second (N = 95) for the confirmatory factor analysis. Table 2 presents demographic characteristics of two groups. The groups were compared using the t-test and chi^2^. They did not differ in terms of the number of participants: x^2^ (1, N = 191) = 0.253; *p* = 0.615; age: t(189) = 0.314, *p* = 0.754, Effect Size d Cohen resp. g Hedges = 0.045; place of residence: x^2^ (3, N = 191) = 2.706; *p* = 0.439; education: x^2^ (3, N = 191) = 4.189; *p* = 0.242; marital status: x^2^ (1, N = 191) = 2.000; *p* = 0.157; child’s disability: x^2^ (1, N = 191) = 0.011; *p* = 0.918.

## 3. Results

### 3.1. Exploratory Factor Analysis for the Polish Version of the PFI

A principal components analysis was conducted using Promax rotation for the 31 items on the Parental Feelings Inventory (PFI). A group of 96 parents described in Table 2 were randomly selected from the group of participants (N = 191). Following the authors of the original version, three-, four-, and five-factor solutions were tested; of these, the three-factor structure yielded the most interpretable results; this was also noted for the original version of the tool. The sample selection (Kaiser, Meyer, and Olkin index) was found to have satisfactory adequacy (0.86).

The items that loaded highly (>0.65) on one of the factors and did not load on more than one factor were retained for the next step; hence, the following items were dropped: afraid, calm, discouraged, grouchy, loving, patient, peaceful, unhappy, and worn out. Unlike the original version, which contains 26 items, 22 items turned out to be significant in the Polish version. The three-factor model consisted of the following factors: *Angry* (angry, annoyed, impatient, irritated, nervous); *Happy* (cheerful, contented, energetic, excited, happy, pleased, relaxed, satisfied), and *Anxious*/*Sad* (frightened, frustrated, guilty, hopeless, miserable, sad, scared, tense, worried). The general factor score for a particular factor was calculated as the sum of the points of all items in the factor divided by the number of items.

### 3.2. Confirmatory Factor Analysis (CFA) for Polish Version of PFI

Then, confirmatory factor analysis (CFA) was performed on the 22 retained items using the remaining 95 parents (49 mothers and 46 fathers, aged 22 to 53, *M* = 37.02; *SD* = 6.82). A three-factor model was fitted with the factors being *Angry*, *Happy*, and *Anxious*/*Sad*. Once again, the adequacy of the sample selection (Kaiser, Meyer, and Olkin index) proved to be satisfactory (0.825). Similar to the original version, although the Chi-square test was significant, χ^2^ (206) = 320.191, *p* < 0.001, the normed Chi-square, χ^2^/df = 1.55, suggested an adequate fit [42]. Other model fit indices also provided support for this three-factor model, e.g., RMSEA = 0.077, SRMR = 0.213. These 22 items and their loadings on respective factors are given in Table 3.

The *Angry* factor was significantly correlated with *Anxious*/*Sad* (0.634; *p* < 0.001) and was significantly negatively correlated with *Happy* (−0.623; *p* < 0.001); in addition, the *Happy* factor was significantly negatively correlated with *Anxious*/*Sad* (−0.488; *p* < 0.01). Factor loadings (individual items and general factor score) were not significantly different between mothers and fathers, i.e., for *Angry*: *t* = 0.024, *p* = 0.981; for *Happy*: *t* = 0.021, *p* = 0.983; for *Anxious*/*Sad*: *t* = −0.078, *p* = 0.938 (Student’s t-test for independent samples), which is consistent with the original version.

### 3.3. Internal Consistency for the Polish Version of the PFI

Cronbach’s *α* coefficients for the total scale slightly differed from those obtained in the original study. Internal consistency for the total scale (22 items) was *α* = 0.78 for mothers and α = 0.76 for fathers. In the original version, Cronbach’s *α* coefficients were slightly higher: *α* = 0.85 for mothers and *α* = 0.87 for fathers. For the Polish version, Cronbach’s *α* coefficients for individual factors were good, with all being above 0.80 (see Table 4). The tool is characterized by satisfactory reliability: all the obtained coefficients exceeded the recommended value of 0.7. Correlation coefficients proved to be high (all above 0.65) between individual questionnaire items and the overall score for the distinguished factors, suggesting that the factors demonstrate high internal consistency.

The external validity of the tool for the mothers and the fathers of healthy children, and those of the children with disabilities, was tested using the t-Student’s test for independent samples. The results are presented in Table 5. Interestingly, although the two groups of parents did not differ with regard to *Angry*, they did differ with regard to *Happy*. Additionally fathers’ scores for *Anxious*/*Sad* are significantly different as well. The size of Cohen’s d indicates the strong effect.

The homogeneity of the variance was assessed with Levene’s test. The results of t-test were interpreted accordingly.

### 3.4. Concurrent Validity for Polish Version of PFI

The concurrent validity of the scale was tested by correlating PFI factor scores with the SPR, SUPIN S20, and SUPIN C20. Our results confirm that the PFI possesses concurrent validity for both mothers and fathers (Table 6). As expected, the *Happy* factor was positively associated with positive affect and positive parental attitudes, and it was negatively associated with negative affect and negative parental attitudes; the factors *Angry* and *Anxious*/*Sad* were positively associated with negative affect and negative parental attitudes, and they were negatively associated with positive affect and positive parental attitudes.

## 4. Discussion

The main purpose of our study was to adapt the original English language version of the Parental Feelings Inventory (PFI) to a Polish context and validate its properties. Firstly, the inventory was translated in the following procedure: translation of the questionnaire text, assessment of translations, back translation, and reaching consensus on the final version of the Polish translation. This was followed by a validation of the internal structure, reliability, and construct validity of the Polish version. The created version of the PFI had acceptable psychometric characteristics; thus, it was found suitable for assessing parental emotions among individuals acting as parents in the Polish population.

Similar to the original version, the Polish PFI has three factors: *Angry*, *Happy*, and *Anxious*/*Sad*; however, it differs from the original version with regard to their composition, i.e., the number of items and their names. The original PFI consisted of 26 items, while 22 items provided the best fit in the Polish version. In the original version, the following terms were dropped from the initial list of 31 items during its design: discouraged, energetic, guilty, tense, and worn out. In the Polish version, the following adjectives were dropped following analysis: afraid, calm, discouraged, grouchy, loving, patient, peaceful, unhappy, and worn out. It can be seen that the original PFI and the Polish version differ in content, but this may be due to the nature of the two languages.

In addition, the general factor score was calculated as the sum of points of all items divided by the number of items. This approach allows a standardized indicator to be obtained for each factor, which takes into account the different number of items in each scale and thus allowing comparisons to be made between factor scores. Our results based on CFA indicate that as with the original version, our Polish adaptation has good construct validity: it demonstrates a well-fitting measurement structure. As in the original, the factor loadings of items and general factor score did not differ between mothers and fathers. The internal and concurrent validity of the scale were also supported. Although the original version demonstrated slightly higher Cronbach’s *α* coefficients than the Polish version, the latter also appears to demonstrate good internal consistency. In addition, our findings indicate that the Polish PFI is characterized by good reliability and internal validity. The factor structure is also consistent with that identified by Bradley et al. [17] and is in line with their conceptual models of general emotion, i.e., with anger, happiness, anxiety, and sadness as basic emotions.

The comparison of PFI scores for parents of non-disabled children and parents of disabled children shows that for women, mothers of non-disabled children differ from those of disabled children with regard to *Happy* but not with regard to *Angry* and *Anxious*/*Sad*. For men, the fathers of non-disabled children differ from those of disabled children with regard to *Happy* and *Anxious*/*Sad* but not *Angry*. Numerous studies show that the parents of typically developing children undergo different emotional experiences to those of children with disabilities [43,44,45]. The differences observed in the levels of *Anxious*/*Sad* feelings are in line with previous studies: Olsson and Hwang [46] found that a number of chronic stressors are inherent in raising a child with a disability, and these may predispose the parents to depression and anxiety. Previous studies also confirm a positive association between parenting a child with disability and the presence of depression and anxiety symptoms [47]. Interestingly, both the mothers and fathers of non-disabled children and those of disabled children do not appear to differ in terms of feeling *Angry*, but they differ in the level of feeling joy. As the existing literature does not include any research related to the study of anger in parenthood, our results are suggestive of further research in this area. Positive emotions have been suggested to play an important role in relation to stressful events, serving to restore coping resources [48], which is relevant in the context of parenting a child with disability. So far, studies of the emotional experiences of parents with children with disability have tended to focus mainly on mothers and on their negative experiences. Our results once again point to a significant area for further verification: positive parental emotions in families with a disabled child and differences in the maternal and paternal approaches to parenting such children.

One of the most important values of our study is that it presents the first tool in Poland to assess specific feelings, i.e., the feelings experienced by mothers and fathers fulfilling the parental role. This fills a gap in existing literature, which usually considers this area as two separate lines of study. So far, research linking the psychology of emotions and family psychology in Poland is extremely rare, and it is usually focused on issues of stress [49,50] and depression [51]; it also typically employs methods for assessing general affect [52]. However, the international literature suggests that parental feelings are extremely specific, and their study requires a specific approach; this need may be met by the PFI, which appears to be an effective tool for assessing parental feelings in both its original and Polish versions.

Our study has some limitations. First, as in the original version, the only measure of reliability is its internal consistency: test–retest data should be collected in future studies to offer further support. Second, our participants were parents of children across a fairly wide age range, i.e., the inclusion criterion was based on having a child under the age of 18, and their total number of children was not taken into account: only parents with more than one disabled child were excluded. Perhaps these factors should be considered in subsequent studies. Third, preserving the semantic meaning of the original version presented a particular challenge, as our goal was to obtain a culturally relevant translation rather than a literal one. However, all discrepancies in translation were thoroughly discussed throughout the process, thus arriving at a final version of the questionnaire containing phrases that most accurately reflected the cultural significance of items. However, the Polish version of the tool differs from the original version in the factor structure. This may be related primarily to linguistic and cultural differences but also to a different group of respondents; in the original study, it was limited to parents of preschool children, while our study concerned parents of children of various ages. Perhaps this resulted in the importance of other parental emotions, which, however, is an advantage of this study: the tool can be used to study the parents of children in a wide range of ages. Finally, it should be emphasized that the PFI, similar to the other methods used in the study, is subject to similar limitations as other self-report measures, as it is based on subjective reflections and recollections.

Although important, the limitations mentioned above do not diminish the presented results, which suggest that the Polish version of the PFI is a valid and reliable tool for measuring parental feelings.

## 5. Conclusions

The adapted version of the PFI was found to be suitable for assessing parental emotions among individuals acting as parents in the Polish population. It is the first such tool for the population of Polish parents in that it measures emotions specifically in the parenting role and combines two extremely important areas of psychology: studies of emotions and studies of family. The results of our research, which include parental emotions as a specific type of emotion, draw attention to the gap in the existing literature and emphasize the need for future research to fill it. The Polish version of PFI will allow for comparative and cross-cultural studies of parenting.

## Figures and Tables

**Table 1 ijerph-18-06003-t001:** Characteristics of the participants (*n* =191).

		Parents of Children without Disabilities		Parents of Children with Disabilities	
		Mothers	Fathers	Mothers	Fathers
Number of Participants		69	59	33	30
Participant age		22–53	22–55	22–52	22–54
		*M* = 35.71	*M* = 37.85	*M* = 37.12	*M* = 39.30
		*SD* = 6.47	*SD* = 6.82	*SD* = 6.95	*SD* = 7.24
Place of residence	village	42	36	19	17
	city (<20,000 residents)	10	9	2	1
	city (20,000–100,000)	10	7	10	10
	city (>100,000 residents)	7	7	2	2
Education	primary	0	2	1	0
	vocational	8	15	7	6
	high school	25	12	9	12
	university	36	30	16	12
Marital status	married	68	58	33	30
	informal relationship	1	1	0	0

**Table 2 ijerph-18-06003-t002:** Comparison of the participants for the exploratory (*n* = 96) and for the confirmatory factor analysis (*n* = 95).

		Group of Participants for the Exploratory Factor Analysis	Group of Participants for the Confirmatory Factor Analysis
Number of participants	mothers	53	49
	fathers	43	46
Participant age		22–55	22–53
		*M* = 37.33	*M* = 37.02
		*SD* = 6.92	*SD* = 6.82
Place of residence	village	60	54
	city (<20,000 residents)	10	12
	city (20,000–100,000)	15	22
	city (>100,000 residents)	11	7
Education	primary	3	0
	vocational	20	16
	high school	30	28
	university	43	51
Marital status	married	94	95
	informal relationship	2	0
Child’s disability	yes	32	31
	no	64	64

**Table 3 ijerph-18-06003-t003:** CFA standardized factor loadings for each item of the PFI for parents (N = 95).

Items [in Polish]	*Angry*	*Happy*	*Anxious*/*Sad*
angry [zły]	0.74		
annoyed [zirytowany]	0.75		
impatient [niecierpliwy]	0.63		
irritated [rozdrażniony]	0.76		
nervous [nerwowy]	0.83		
cheerful [radosny]		0.93	
contented [zadowolony]		0.86	
energetic [pełen energii]		0.87	
excited [podekscytowany]		0.83	
happy [szczęśliwy]		0.93	
pleased [spełniony]		0.85	
relaxed [zrelaksowany]		0.79	
satisfied [usatysfakcjonowany]		0.74	
frightened [przerażony]			0.78
frustrated [sfrustrowany]			0.86
guilty [winny]			0.70
hopeless [bezradny]			0.87
miserable [przygnębiony]			0.89
sad [smutny]			0.72
scared [przestraszony]			0.83
tense [spięty]			0.88
worried [zmartwiony]			0.71

**Table 4 ijerph-18-06003-t004:** Correlation coefficients of items with the general factor score and Cronbach’s α coefficients (N = 95).

Measures	*Angry*	*Happy*	*Anxious*/*Sad*
**Mean (SD)**	2.98 (1.13)	4.61 (1.25)	2.59 (1.02)
**Items [in Polish]**	*r*	*r*	*r*
angry [zły]	0.80 ***		
annoyed [zirytowany]	0.66 ***		
impatient [niecierpliwy]	0.76 ***		
irritated [rozdrażniony]	0.82 ***		
nervous [nerwowy]	0.80 ***		
cheerful [radosny]		0.83 ***	
contented [zadowolony]		0.83 ***	
energetic [pełen energii]		0.81 ***	
excited [podekscytowany]		0.73 ***	
happy [szczęśliwy]		0.82 ***	
pleased [spełniony]		0.84 ***	
relaxed [zrelaksowany]		0.71 ***	
satisfied [usatysfakcjonowany]		0.79 ***	
frightened [przerażony]			0.72 ***
frustrated [sfrustrowany]			0.74 ***
guilty [winny]			0.68 ***
hopeless [bezradny]			0.69 ***
miserable [przygnębiony]			0.72 ***
sad [smutny]			0.67 ***
scared [przestraszony]			0.76 ***
tense [spięty]			0.75 ***
worried [zmartwiony]			0.74 ***
**Cronbach’s alpha**			
Mothers	0.84	0.92	0.89
Fathers	0.81	0.93	0.87

*** *p* < 0.001.

**Table 5 ijerph-18-06003-t005:** Comparison of PFI scores for parents of non-disabled children and parents of disabled children (N = 95).

		Mean (SD)		*t*	*p*	*df*	Effect Size d Cohen resp. g Hedges
		Non-Disabled Child	Disabled Child				
Mothers	*Angry*	2.98 (1.98)	2.99 (1.38)	0.22	0.983	47	0.008
	*Happy*	4.66 (0.94)	3.84 (1.27)	−2.51	0.016	47	0.787
	*Anxious*/*Sad*	2.43 (1.03)	3.00 (1.23)	1.56	0.126	47	0.496
Fathers	*Angry*	2.79 (0.98)	3.29 (1.09)	1.63	0.110	44	0.490
	*Happy*	4.72 (1.01)	3.91 (0.89)	−2.74	0.009	44	0.837
	*Anxious*/*Sad*	2.27 (0.74)	3.16 (0.94)	3.55	0.001	44	0.779

*t*—t-Student’s test.

**Table 6 ijerph-18-06003-t006:** Correlations between PFI factors and variables of interest for mothers and fathers (N = 95).

	**Mothers**			**Fathers**		
Measures	***Angry***	***Happy***	***Anxious*/*Sad***	***Angry***	***Happy***	***Anxious*/*Sad***
Mean (SD)	2.98 (1.23)	4.43 (1.10)	2.59 (1.11)	2.97 (1.04)	4.42 (1.04)	2.60 (0.92)
	*r*	*r*	*r*	*r*	*r*	*r*
SPR						
accepting attitude	−0.18	0.45 **	−0.21	−0.47 **	0.43 **	−0.28 *
overly demanding attitude	0.02	−0.15	0.14	0.04	0.19	−0.14
autonomous attitude	−0.18	0.47 **	−0.20	−0.04	0.34 *	−0.35 *
inconsistent attitude	0.62 ***	−0.29 *	0.37 **	0.31 *	−0.15	0.03
excessively protective attitude	−0.09	−0.03	0.15	−0.36 *	−0.03	0.01
SUPIN S20 “now”						
positive affect	−0.23	0.33 *	−0.16	−0.20	0.54 **	−0.35 *
negative affect	0.26	−0.24	0.43 **	−0.05	−0.36 *	0.21
SUPIN C20 “usually”						
positive affect	−0.14	0.30 *	−0.21	−0.11	0.38 *	−0.37 *
negative affect	0.46 **	−0.23	0.50 ***	0.31*	−0.40 **	0.40 **

* *p* < 0.05; ** *p* < 0.01; *** *p* < 0.001.

## Data Availability

The data is available at https://osf.io (accessed date: 1 June 2021) (databases: https://osf.io/59sym; https://osf.io/3aubx; https://osf.io/9w2gx; syntax: https://osf.io/rq5ud (accessed date: 1 June 2021)). The code book is available at https://osf.io/7z2ju (accessed date: 1 June 2021).

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
