# Peer review of "Psychometric Properties of the Polish Version of the Parental Feelings Inventory"

_ijerph, 2021, doi:10.3390/ijerph18116003_

Round 1

Reviewer 1 Report

This is an important contribution to the literature and has scientific merit and is overall well written. 

However, a couple of issues exist with writing and grammar.  The first sentence of the abstract could be reworded to be more succinct and clear. In addition, on line 15, page 1, it is not clear how one would "adapt" psychometric properties?

Lines 90-91 of page two misrepresents the method that Bradley et al. used in survey development. The terms emotions, mood, and affect were not examined in the literature as part of survey development. This section should be rewritten.

On page 6, line 243, the authors state that “the two groups did not differ in terms of demographic characteristics (Table 2).” However, I don’t see any statistical evidence to support this assertion and it should be included. Eyeballing demographic differences is not sufficient.

Reasons for the differences in the English and Polish versions should be explored further. For example, in what ways do the two languages differ and how might those differences have led to different factor structures?

Author Response

Dear Reviewer,

We are grateful for your time and constructive comments on our manuscript. We have implemented comments and suggestions and wish to submit a revised version of the manuscript for further consideration in the journal. Changes in the initial version of the manuscript are highlighted in the revised version. Below (please see the attachment), we also provide a point-by-point response explaining how we have addressed each of the comments. We look forward to the outcome of your assessment.

Yours sincerely,

Dominika Fijałkowska and Eleonora Bielawska-Batorowicz

Reviewer 2 Report

I'm sorry, but the wording of my English isn't excellent. If something is not understood, please write to me asking for clarification.   I haven't detected any errors. My comments are suggestions for improvement, so that the study has more impact and is more cited.   Table 1. Delete the last row (Child's disability)   Tables 2 and 3 add the item in Polish   Demonstrated reliability and carried out the construct validity study with the Confirmatory Factorial Analysis, it is logical to include the entire sample for the results of Tables 5 and 6. I propose to update the calculations with the entire sample for Tables 5 and 6 (the significant results will possibly increase).   Table 5. I find SD 2.98 strange, very different from the rest. Verify that there is no error in the data. In the calculation of the t-test, has the homogeneity of variances been taken into account? (SPSS provides a correction for the t-test).   I consider that table 5 should add the significance obtained in each test, as well as the size of the effect (e.g. Cohen D with Hedges correction). You do not need to include the gl data.   It's a good article, coherence between theory, study and results discussion.   Congratulations  

Author Response

Dear Reviewer,

We are grateful for your time and positive and constructive comments on our manuscript. We have implemented comments and suggestions and wish to submit a revised version of the manuscript for further consideration in the journal. Changes in the initial version of the manuscript are highlighted in the revised version. Below (please see the attachment), we also provide a point-by-point response explaining how we have addressed each of the comments. We look forward to the outcome of your assessment.

Yours sincerely,

Dominika Fijałkowska and Eleonora Bielawska-Batorowicz
